# Printing a Pacinian Corpuscle: Modeling and Performance

**DOI:** 10.3390/mi12050574

**Published:** 2021-05-18

**Authors:** Kieran Barrett-Snyder, Susan Lane, Nathan Lazarus, W. C. Kirkpatrick Alberts, Brendan Hanrahan

**Affiliations:** 1Sensors and Electron Devices Directorate, U.S. Army Research Laboratory, Adelphi, MD 20783, USA; kbarret1@ucsc.edu (K.B.-S.); sel8819@rit.edu (S.L.); nathan.lazarus2.civ@mail.mil (N.L.); william.c.alberts4.civ@mail.mil (W.C.K.A.II); 2Electrical and Computer Engineering, University of California, Santa Cruz, CA 95064, USA

**Keywords:** 3D printing, soft robotics, vibration sensing, biomimetic

## Abstract

The Pacinian corpuscle is a highly sensitive mammalian sensor cell that exhibits a unique band-pass sensitivity to vibrations. The cell achieves this band-pass response through the use of 20 to 70 elastic layers entrapping layers of viscous fluid. This paper develops and explores a scalable mechanical model of the Pacinian corpuscle and uses the model to predict the response of synthetic corpuscles, which could be the basis for future vibration sensors. The −3dB point of the biological cell is accurately mimicked using the geometries and materials available with off-the-shelf 3D printers. The artificial corpuscles here are constructed using uncured photoresist within structures printed in a commercial stereolithography (SLA) 3D printer, allowing the creation of trapped fluid layers analogous to the biological cell. Multi-layer artificial Pacinian corpuscles are vibration tested over the range of 20–3000 Hz and the response is in good agreement with the model.

## 1. Introduction

When you run your fingers along a surface, interactions between your fingerprints and surface topography generate vibrations in your skin [1]. These vibrations are picked up by mechanoreceptors and encoded as neural stimuli. The Pacinian Corpuscle (PC) is the most sensitive mechanoreceptor in the human body in the range of 5 Hz–1 kHz [2,3,4,5] with a peak sensitivity at 250 Hz [6,7], and is capable of sensing vibrations with a minimum displacement of 10 nm [8]. This high sensitivity gives humans their fine sense of touch [9] and the corpuscle’s sensitivity range results from the cell’s complex multi-layered structure [10,11,12]. Another function of the PC is found in elephants, where the cell is thought to be used in the detection of seismic waves [13], allowing for communication between elephants as well as for the detection of potential threats [14]. An artificial sensor based on the PC could find uses in tactile interfaces for medical instruments or prosthetics [15,16,17] and seismic vibration sensing for perimeter monitoring [18].

The PC has been the subject of many studies due to its large size, reaching lengths of up to 4 mm, making it easier to observe than other mechanoreceptors [19], and filtering properties. There have been two primary approaches to creating a model that describes the filtering: functional [20,21,22,23] and mechanical [10,24,25]. A functional approach uses a transfer function fit to experimental data. This approach has the advantage of being able to accurately capture the behavior of the PC. However, functional models fail to describe the mechanics of the PC. A functional model is useful for modeling a known PC, but is less helpful for predicting the behavior of an artificial sensor based on the same principles. A mechanical model, on the other hand, describes the behavior of the PC based on material and structural properties. This type of model is more useful for predicting how new artificial PCs would behave, and how changes in properties affect the sensor behavior.

Loewenstein and Skalak were among the first to create a mechanical model of the PC [10]. However, the Loewenstein–Skalak (LS) model was relatively simple and neglected both the mechanics of the core of the PC and the effects of mass on the filter. The LS model was also found to have other limitations, including deriving material properties from the properties of the arterial walls, which later studies found do not match well [26,27,28], and difficulty scaling to Pacinian corpuscles of different sizes.

A later model by Biswas sought to overcome the limitations of the LS model. This model uses more accurate parameters than had previously been approximated. Additionally, they considered the core as another viscoelastic layer and made the model more scalable. The limitations of this model include the use of approximation for the transfer function to save computation time [29]. Notably, the modeling of Pacinian corpuscles has focused on understanding the biological system, and while there have been attempts at creating an artificial PC in the past [30], to our knowledge, there has been no attempt to model and design artificial Pacinian corpuscles through a full fluid–solid mechanical model.

Here, we present the first fluid–solid mechanical model of an artificial PC. The artificial PC here is 3D printed using stereolithography, allowing for complex, multi-layer geometries. The fluid pockets fundamental to the Pacinian corpuscle are integrated during the SLA printing process through fluid entrapment of uncured photoresin, an approach previously demonstrated for self-healing soft robotic structures in [31]. A small, sensitive vibration sensor, such as the PC, functioning similarly to ones existing in nature, can find uses ranging from tactile interfaces in robotics to nonvisual detection of enemy movement in military scenarios. The advantage of a sensor based on the PC is its passive filtering mechanics, saving time and resources on signal processing.

## 2. Theory

The PC is composed of three major parts, the nerve fiber, inner core, and perineural capsule. The nerve fiber is the site of mechanotransduction in the PC, containing many stretch-activated ion channels that convert mechanical stimulus into electrical spikes [32]. The inner core consists of many densely packed lamellae that surround the nerve fiber [33]. The perineural capsule of the PC consists of 13–70 concentric layers of a viscous fluid trapped by a flexible membrane (lamella). The spacing between adjacent layers is maintained by randomly distributed solid connections [19,26,31,34,35,36,37]. The main role of the perineural capsule is to filter out low-frequency vibrations. In response to pressure applied on an axis orthogonal to the neuron, the outermost lamella attempts to compress along the axis of applied pressure. Because the fluid is incompressible, the sides of the lamella perpendicular to the compression must expand to keep the total fluid volume constant. When pressure is applied slowly, the fluid is able to flow from the areas of compression to those of expansion without disturbing the inner layers. When pressure is applied quickly, the fluid flow is restricted by the viscosity, and the pressure applied on the lamella is transmitted through the fluid to the next layer [7,10,11].

When describing the behavior of each layer, the following characteristics must be accounted for: the stiffness of the lamella, the viscous fluid resistance, the stiffness of the interlamellar spring matrix, and the mass of the entire layer. The equations were originally derived by [10] and later modified by [24]. These parameters are used to calculate the previously mentioned layer characteristics.

Lamellar stiffness, *K_L_*, is modeled by a simply supported cylindrical membrane based on the Timoshenko beam theory [24].
$$K_{L}=\frac{E_{L}AT}{R{\left(1+\frac{4L^{2}}{\pi^{2}R^{2}}\right)}^{2}}$$

Viscous resistance, *B*, follows lubrication theory [24].
$$B=\frac{12\mu L^{2}A}{\pi^{2}H\left(1+\frac{4L^{2}}{\pi^{2}R^{2}}\right)}$$

Interlamellar matrix stiffness, *K_M_*, follows a linear elastic model [24].
$$K_{M}=\frac{E_{M}A}{H}$$

## 3. Design and Simulation

The selection of the material parameters for the Pacinian corpuscle has changed over the years. Loewenstein, et al. assumed the elastic modulus of the lamella to be 500 kPa, which is on the order of tissue such as arterial walls, and the fluid viscosity to be the same as water, 1 mPa∙s [10]. The elastic modulus of the interlamellar matrix was used as a fitting parameter, which was determined to best fit the experimental data of [11] at 50 Pa. The later model by Biswas et al. uses an elastic modulus of 1 kPa, based on the specific collagen that makes up the lamellae [26,38], and a fluid viscosity between 1.4 and 7 mPa∙s due to collagen fibers present in the fluid [28]. For the interlamellar matrix, 1 Pa was used [24]. Later, Quindlen et al. used micropipette aspiration to measure the apparent elastic modulus of the entire PC. From their study, a value of 1.4 ± 0.86 kPa was obtained [27], which validates the assumption used by Biswas.

The Pacinian corpuscle model consists of *N* concentric cylindrical layers stacked on top of each other. Each layer is represented by a Kelvin–Voigt model with an added mass and parallel spring. The spring and damper in the Kelvin–Voigt model represent the interlamellar matrix stiffness and the viscous fluid resistance, respectively, and the parallel spring is the lamellar stiffness (Figure 1b). The inner core is represented by a Kelvin–Voigt model with added mass and a series spring (Figure 1c) [39]. The derivation of the components of the layers originates from Loewenstein [10], who assumes that the core is rigid. The derivation of the core components is performed separately by Biswas [29], who reuses the derivations of the layer components from the Loewenstein model.

In creating an artificial PC, understanding the behavior of this geometry for printable layers of available engineering materials is necessary. The complex compliance, *G(s)*, of layer *i* can be calculated using the equation below, where *s* is the complex frequency. The subscripts *s* and *p* refer to the series and parallel stiffness coefficients for the core (*c*), while the subscripts *M* and *L* refer to the matrix and lamellar stiffness, respectively:$$G_{i}\left(s\right)=\frac{X_{i}\left(s\right)}{F_{i}\left(s\right)}=\frac{1}{K_{Li}+\frac{1}{G_{si}\left(s\right)+G_{si-1}\left(s\right)}}$$
where:$$G_{si}\left(s\right)=\frac{1}{M_{i}s^{2}+B_{si}s+K_{Mi}}$$

The core is considered layer 0, and has a complex compliance of the form:$$G_{0}\left(s\right)=\frac{X_{c}\left(s\right)}{F_{c}\left(s\right)}=\frac{1}{K_{sc}}+\frac{1}{M_{c}s^{2}+B_{sc}s+K_{Pc}}$$

The transmittance of compression between layers was derived based on these compliances:$$\frac{X_{i-1}\left(s\right)}{X_{i}\left(s\right)}=\frac{G_{i-1}\left(s\right)}{G_{si}\left(s\right)+G_{i-1}\left(s\right)}$$

The compression on the core in response to the compression on the outermost layer is found iteratively, where *m* is the product of *p*i* from 1 to *N*.
$$\frac{X_{0}\left(s\right)}{X_{N}\left(s\right)}=\overset{N}{\underset{m=1}{\prod }}\frac{G_{m-1}\left(s\right)}{G_{m-1}\left(s\right)+G_{sm}\left(s\right)}$$

The model was coded in MATLAB and simulated for varying fluid viscosity, material stiffness, number of layers, and the inclusion of the interlamellar matrix (Figure 2).

In the pass band of the PC (5 Hz–1 kHz), a sharp increase in transmittance magnitude is observable. This increase falls off above 1 kHz, which is at the high end of the PC’s sensitivity range.

Changes in the viscosity result in an inverse change in the 3 dB point. The connection between lubrication theory, viscosity, and response is as follows: At low frequencies and/or low viscosity, the fluid is able to re-distribute within the lamella and outside perturbations are not transmitted to the core. As the fluid becomes more viscous or at higher frequencies, the fluid cannot redistribute and displacements are translated to the core. Increasing the elastic modulus results in an increased 3 dB point. Increasing the number of layers of a PC increases its 3 dB point. Adding layers increases the filtering capabilities of the PC. The effective stiffness of the fluid must exceed the stiffness of the layers and core in order to transmit compression effectively. Additionally, the effect of removing the interlamellar matrix is observed. For static and very low-frequency compression, the compression transmittance is increased when the matrix is included. However, in the functional range of the PC, there is no significant effect on compression transmittance. Through these parametric sweeps, it is noted that the filtering effect is most sensitive to changes in the viscosity of the fluid layers.

## 4. Results and Discussion

To validate the accuracy of the model, the testing of an artificial PC is performed. First, a modified PC is modeled based on the restrictions of our fabrication process (including size and material constraints), and PCs with similar behavior to the one in Figure 2a are created. Then, the artificial PCs are subjected to indentation to determine compression transmittance at varying frequencies.

### 4.1. Fabrication

The artificial PC capsules are created using a Formlabs Form 2 stereolithographic (SLA) 3D printer. The SLA technique is based on layer-by-layer light-induced curing of a resin. Using an SLA printer, it is possible to create structures that contain uncured liquid resin sealed inside from the printing process [40]. This capability is ideal for simultaneously realizing the liquid and solid components of complex structures. A photograph of the artificial PC is shown in Figure 3 with part of the outer layer removed to reveal the inner geometry.

Table 1 compares the material properties of the PC with those of the 3D-printer material. Given the differences in some mechanical properties and the geometrical limitations of the SLA printer, some alterations compared to a biological PC are needed. Despite these structural differences, a PC that has fewer but thicker layers can produce a similar frequency response to one that has more but thinner layers [41].

Figure 4 shows the simulated comparison between the modeled PC from the previous section and a five-layer 3D-printable design. The high-frequency jitter seen in the range of 5 to 7 kHz is attributable to the mass resonance of the core. A higher transmission for static compression is caused by fewer layers in the system. Other than these factors, the printable model behaves similarly to the model PC. Moving forward, artificial PCs with one to five layers are 3D printed.

### 4.2. Experiments

Vibration testing of the filter is performed with a shaker table (Ling Dynamics V408). The setup is shown in Figure 5. The artificial PC is mounted to a static top plate. The stiffness of the restraint is minimal when compared to the stiffness of the lamella and can be ignored. The shaker applies an input displacement to the bottom through a smooth point load to focus the input to the center of the corpuscle and minimize the mechanical influence from the ends of the cylinder. Two-layer and three-layer corpuscles were selected for vibration testing. Sinusoidal vibrations were imparted on the sensor at frequencies ranging from 20 to 3000 Hz.

A 3 mm diameter hole was made lengthwise through the core in order to feed a piezoelectric cable (CAT-PFS0002) through the center to act as the sensor for the output compression. The piezoelectric sensor was chosen because, similar to the nerve fiber in the actual PC, piezoelectric materials output electrical signals in response to applied strain. The signal from the piezoelectric cable was conditioned by a charge amplifier with a gain of 10^9^ and a minimum frequency of 7 Hz. This frequency is selected to be out of the range of interest for testing.

Input displacement was measured by a co-located PCB Piezotronics 3-axis accelerometer (Model 356A12). The lowest displacement measured during testing is around 15 nm.

In Figure 6, the two and three-layer PCs show an increase in compression transmittance in the range of 20 Hz–1 kHz with a 3 dB point around 1.3 kHz. This corresponds well with the model, which predicts a similar response for these PCs. The two-layer PCs have a lower 3 dB point than the three-layer ones, which is also supported by the model, as fewer layers produce a weaker filtering effect. Some non-linear responses are noticeable at frequencies that are a multiple of 60 Hz. These irregularities are caused by the 60 Hz electrical noise coming from the piezo cable. The response of the artificial PC could not be determined for frequencies below 20 Hz, but these frequencies are not in the range of interest for this paper.

## 5. Conclusions

In this paper, a tunable model of the behavior of the layered structure of the Pacinian corpuscle was described and used to design a 3D printed artificial PC. The artificial PC exhibited the passive mechanical filter response matching the one predicted by the model. The model is limited to small displacements of the lamella, assuming that each layer will not make contact with another. Additionally, the model assumes homogeneity throughout the layer. The artificial PCs created are entirely homogeneous in construction and testing was limited to small indentations in compliance with these limitations. The artificial PCs created were also three orders of magnitude larger and stiffer than real PCs. An improvement in the printing resolution to allow a smaller PC or the use of a more compliant 3D printing material would enable the fabrication of synthetic PCs more closely resembling the real PC. Another consideration for operating at lower frequencies is the internal sensor that feeds through the center. In this paper, a piezoelectric cable is used for its linear strain–charge response in a specific frequency band. Piezoelectrics are less suitable for low frequency and especially static strains. Another sensor better suited for low-frequency strains may be considered for future work.

## Figures and Tables

**Figure 1 micromachines-12-00574-f001:**
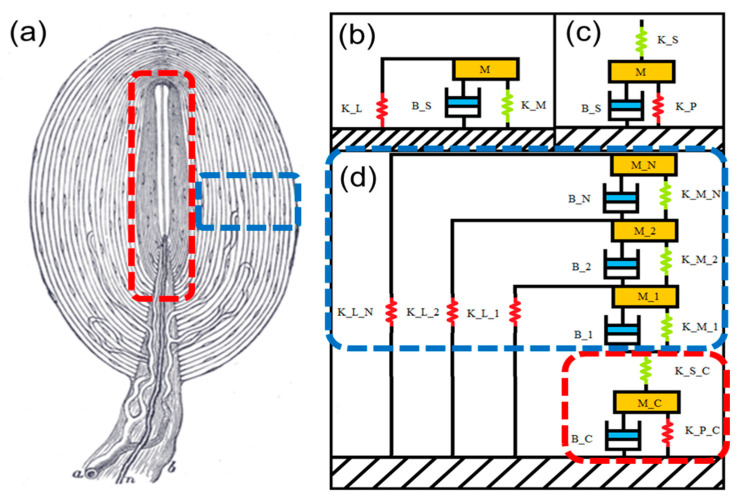
(**a**) Diagram of the PC structure, modified from [40]. (**b**) Mechanical model of a single layer, Kelvin–Voigt with a parallel spring and mass. (**c**) Mechanical model of the core, Kelvin–Voigt with a series spring and mass. (**d**) Mechanical model of an entire corpuscle.

**Figure 2 micromachines-12-00574-f002:**
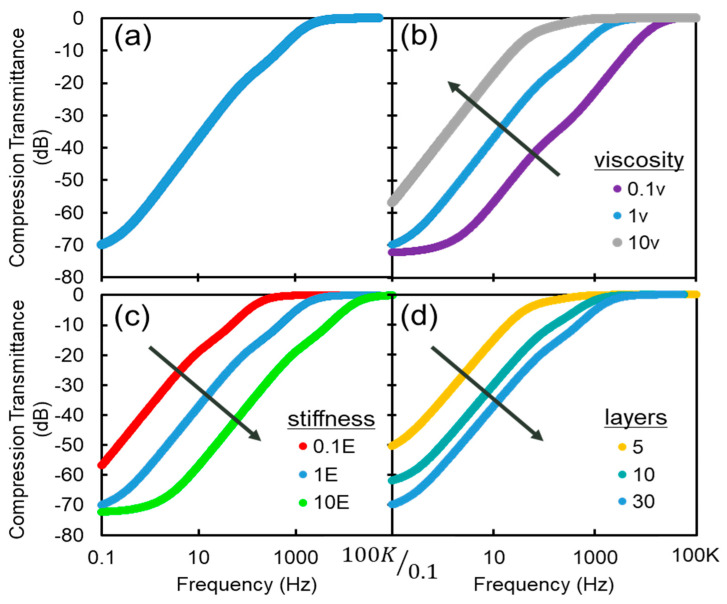
(**a**) Frequency plot for compression transmittance of a typical PC; (**b**) Parametric sweep of viscosity; (**c**) Parametric sweep of Young’s modulus; (**d**) Parametric sweep of number of layers.

**Figure 3 micromachines-12-00574-f003:**
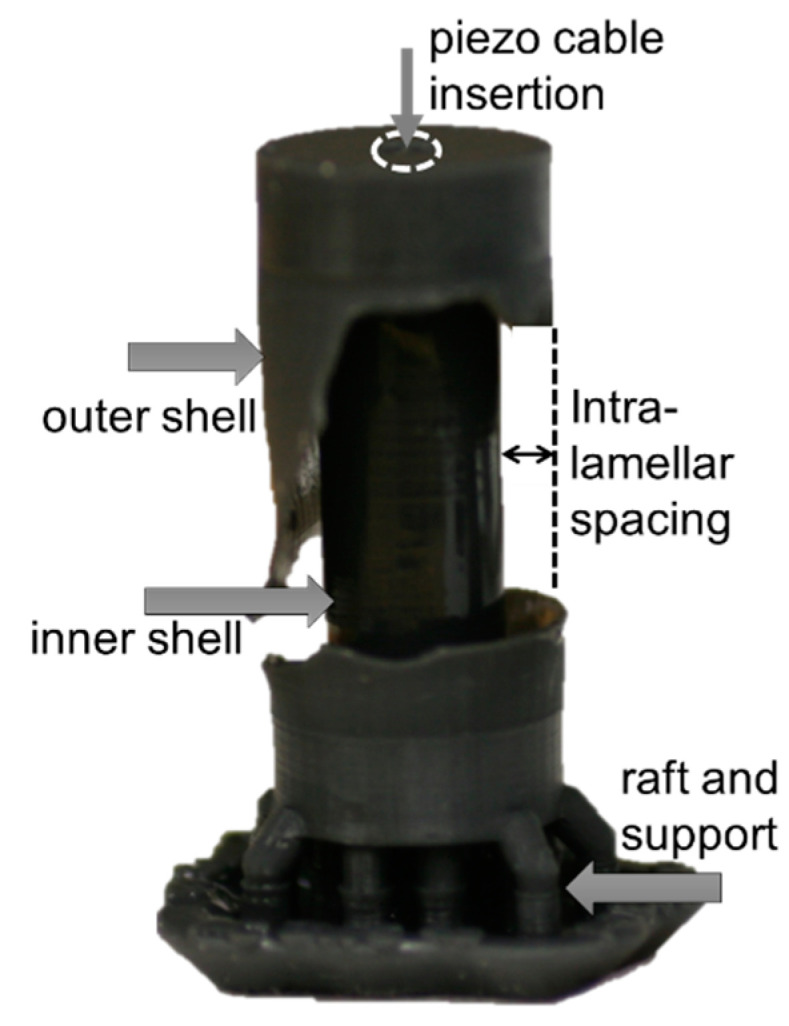
Photograph of the printed artificial Pacinian corpuscle 2-layer design. The outer layer was removed to show the first fluid encapsulation volume.

**Figure 4 micromachines-12-00574-f004:**
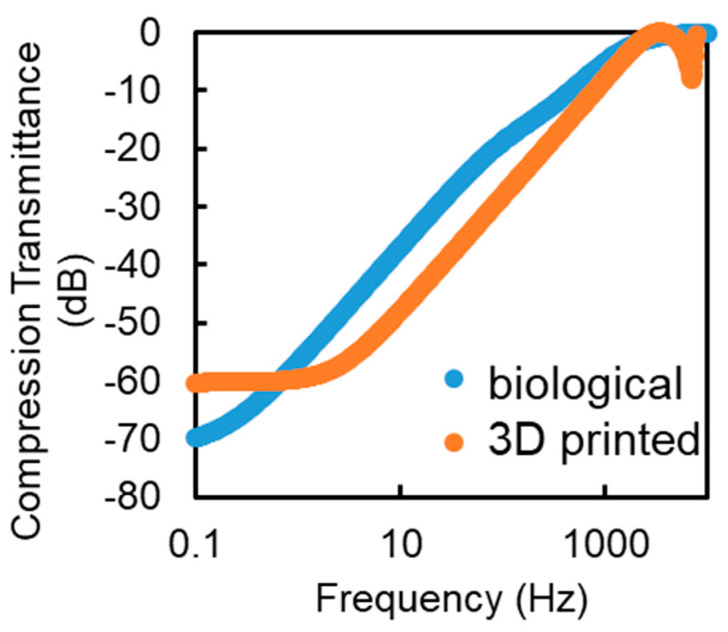
Simulation results for compression transmittance of a typical PC (blue) compared with a 5 layer 3D printable PC (orange).

**Figure 5 micromachines-12-00574-f005:**
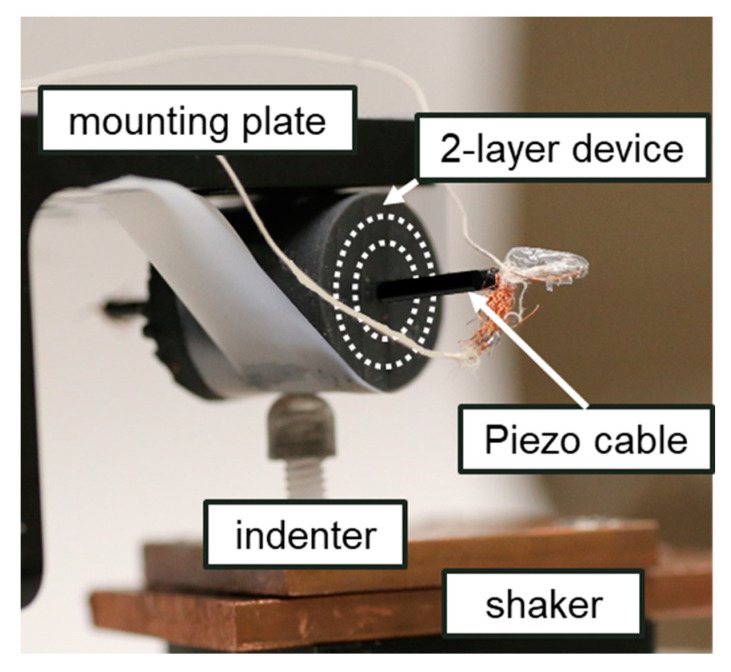
Experimental setup for vibration testing. Accelerometer not pictured. The artificial PC is attached to the static top plate and subject to indentation from below.

**Figure 6 micromachines-12-00574-f006:**
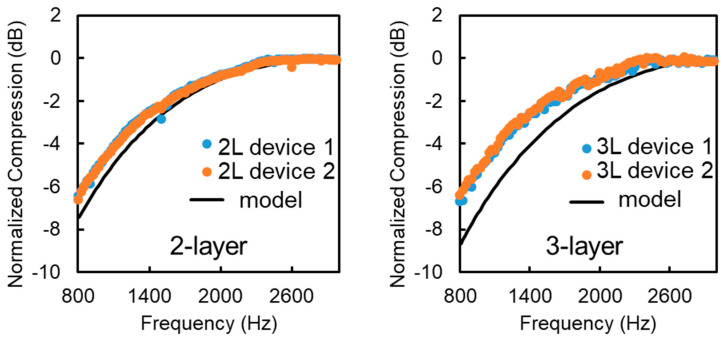
Graph of normalized compression of 2 (**left**) and 3 (**right**) layer artificial Pacinian Corpuscles in the range of 800–3000 Hz. The black line represents the expected compression from the model while the different colored dots represent different corpuscles with the same number of layers.

**Table 1 micromachines-12-00574-t001:** Comparison of the parameters of the real Pacinian corpuscle with the artificial design.

Parameter	Pacinian Corpuscle [24]	3D-Printed Design
Elastic Modulus of Lamella	1 kPa	5.5 MPa
Fluid Viscosity	3 mPa∙s	4.5 Pa∙s
Lamellar Thickness	0.1–0.4 µm	1 mm
Interlamellar Spacing	1–8 µm	2 mm
Core Radius	15 µm	4 mm
Length	1 mm	2 cm
Number of Layers	20–60	1–5

## Data Availability

The data supporting reported results are available upon reasonable request.

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
