# Peer review of "Printing a Pacinian Corpuscle: Modeling and Performance"

_micromachines, 2021, doi:10.3390/mi12050574_

Round 1

Reviewer 1 Report

The manuscript describes the simulation and 3D printing of a Pacinian Corpuscle (PC)-like structure. The simulation indicates that the number of layers, stiffness, and liquid viscosity can influence the compression transmission profiles. 3D printing is based on Form 2 stereolithographic techniques and photocurable resin. It is efficient to use uncured resin to mimic the interlamellar fluid, which is compatible with the 3D printing process. The authors compared natural PC and the 3D printed counterpart in terms of dimension, materials property, and performance. The authors also indicated several things that may be improved in the future, including piezoelectric cable, layer numbers, and printing resolution. Some comments are shown below:

  • It is recommended to characterize the printed structure and the printing performance. There is only one picture to show the printed PC.
  • The theory description seems incomplete. For example, the “M” and “Mcs2” are not clearly defined.  
  • The authors adopted a lubrication theory to describe the viscous resistance. It seems the viscous resistance is irrelevant to the frequency. Please provide discussions on this point.
  • Ref 16 seems incomplete.

Author Response

Review Responses:

Thank you for the advice on how to improve the paper. We have made improvements throughout the manuscript.

R1:

We have added a new figure 3 and some description which shows the as-printed part. More of the details of the printed corpuscle can be viewed in the image.

We have updated our theory section with the appropriate definitions. This was an oversight on our part.

We have added the following discussion:

The connection between lubrication theory, viscosity and response is as follows: At low frequencies and/or low viscosity, the fluid is able to re-distribute within the lamella and outside perrturbations are not transmitted to the core. As the fluid becomes more viscous or at higher frequencies, the fluid cannot redistribute and displacements get translated to the core.

R2:

We have added an image of the printed part as Figure 3.

R3:

We have updated our theory section with the appropriate definitions. This was an oversight on our part.

We have also updated the references to the MDPI ACS style.

Reviewer 2 Report

Since the author has used 3D printing to fabricate the model. An image of the prototype 3D printed model or supplementary video is needed

Author Response

(The authors gave the same response as above.)

Reviewer 3 Report

In this manuscript, Barrett-Snyder and his co-workers present the development of the first scalable fluid-solid mechanical model of the Pacinian corpuscle, as well as the application of this model to predict the response of synthetic corpuscles. They also present the construction of artificial Pacinian corpuscles using uncured photoresist within structures printed in a commercial stereolithography 3D printer. The manuscript, in general, is well-written and well-illustrated. The topic of the paper fits the scope of Micromachines. Results deserve publication.

Minor comments:

--- Please provide the meaning of symbols in equations in lines 99, 101 and 103. Similarly, for equations in lines 133, 135, 137, 141 and 145, if not provided earlier.

--- References should be described as follows (please check journal’s home page):

Journal Articles: Author 1, A.B.; Author 2, C.D. Title of the article. Abbreviated Journal Name Year, Volume, page range.

Books: Author 1, A.; Author 2, B. Book Title, 3rd ed.; Publisher: Publisher Location, Country, Year; pp. 154–196.

Book Chapter: Author 1, A.; Author 2, B. Title of the chapter. In Book Title, 2nd ed.; Editor 1, A., Editor 2, B., Eds.; Publisher: Publisher Location, Country, Year; Volume 3, pp. 154–196.

--- Ref. [16] and [32]: use initials of given names.

--- In most of the references: family name should be followed by initials of given names (not only for the first author).

Author Response

(The authors gave the same response as above.)
